Regulation of soldier caste differentiation by microRNAs in Formosan subterranean termite (Coptotermes formosanus Shiraki)

Du He 1
Huang Runmei 1
Chen Da-Song 1
Zhuang Tianyong 1
Huang Xueyi 1
Zhang Huan 2
Li Zhiqiang 1 lizhiqiang61@163.com
1 Guangdong Key Laboratory of Integrated Pest Management in Agriculture, Guangdong Public Laboratory of Wild Animal Conservation and Utilization, Institute of Zoology, Guangdong Academy of Sciences , Guangzhou , China
2 State Key Laboratory of Integrated Management of Pest Insects and Rodents, Institute of Zoology, Chinese Academy of Sciences , Beijing , China
Lazzari Claudio
Electronic publication date: 2024 Feb 29
Publication date: 2024
Volume: 12
Electronic Location ID: e16843
Received 2023 Jun 1; Accepted 2024 Jan 5
Copyright: © 2024 Du et al.
Copyright year: 2024
Copyright holder: Du et al.
License: This is an open access article distributed under the terms of the Creative Commons Attribution License, which permits unrestricted use, distribution, reproduction and adaptation in any medium and for any purpose provided that it is properly attributed. For attribution, the original author(s), title, publication source (PeerJ) and either DOI or URL of the article must be cited.
License URL: https://creativecommons.org/licenses/by/4.0/

Keywords: Juvenile hormone, Polyphenism, Post-transcriptional regulation, Noncoding RNA

Funding: National Natural Science Foundation 31702068 GDAS Project of Science and Technology Development 2019GDASYL-0104018 State Key Laboratory of Integrated Management of Pest Insects and Rodents, IOZ-CAS IPM2211 GIZ grant GIABR-gjrc201601 This research was funded by the National Natural Science Foundation of China (Grant No. 31702068), GDAS Project of Science and Technology Development (Grant No. 2019GDASYL-0104018), Project of State Key Laboratory of Integrated Management of Pest Insects and Rodents, IOZ-CAS (IPM2211), and GIZ grant (GIABR-gjrc201601). The funders had no role in study design, data collection and analysis, decision to publish, or preparation of the manuscript.

==============================
The soldier caste is one of the most distinguished castes inside the termite colony. The mechanism of soldier caste differentiation has mainly been studied at the transcriptional level, but the function of microRNAs (miRNAs) in soldier caste differentiation is seldom studied. In this study, the workers of Coptotermes formosanus Shiraki were treated with methoprene, a juvenile hormone analog which can induce workers to transform into soldiers. The miRNomes of the methoprene-treated workers and the controls were sequenced. Then, the differentially expressed miRNAs (DEmiRs) were corrected with the differentially expressed genes DEGs to construct the DEmiR-DEG regulatory network. Afterwards, the DEmiR-regulated DEGs were subjected to GO enrichment and KEGG enrichment analysis. A total of 1,324 miRNAs were identified, among which 116 miRNAs were screened as DEmiRs between the methoprene-treated group and the control group. A total of 4,433 DEmiR-DEG pairs were obtained. No GO term was recognized as significant in the cellular component, molecular function, or biological process categories. The KEGG enrichment analysis of the DEmiR-regulated DEGs showed that the ribosome biogenesis in eukaryotes and circadian rhythm-fly pathways were enriched. This study demonstrates that DEmiRs and DEGs form a complex network regulating soldier caste differentiation in termites.

Introduction

Termites are a typical example of polyphenism, and individuals with divergent morphologies are categorized into different castes in the colony (Nijhout, 1999; Simpson, Sword & Lo, 2011; Wilson, 1971). Among these castes, the soldier caste is responsible for defending the colony. The ratio of the soldier caste is relatively stable, which is a characteristic of each termite species (Haverty, 1977). Termites in other castes can develop into soldiers when the soldier ratio falls below the standard level in the colony. The soldier caste develops differently in different species; for example, the soldier caste develops from pseudergates in Kalotermes, while in Termitidae it is larvae that develop into soldiers (Edwards & Mill, 1986). Before a termite becomes a soldier, it experiences a presoldier stage, and then molts again into the soldier stage. This transformation is controlled by both the termite’s genomic information and the social environment inside the colony (Miura, 2005; Noirot, 1991). One key substance that drives the transformation into soldiers is juvenile hormone (JH; Korb, 2015; Lüscher, 1969; Lüscher & Springhetti, 1960). Consistently high JH titer is required for transformation into the presoldier stage (Cornette et al., 2008; Nijhout & Wheeler, 1982); therefore, the application of juvenile hormone analog (JHA) has been shown to induce the change to presoldier in termites (Hrdý & Křeček, 1972; Su & Scheffrahn, 1990).

The most obvious change is the development of fat body at the histological level during the transformation from worker to soldier in Hodotermopsis sjostedti Holmgren (Cornette, Matsumoto & Miura, 2007). During this process, the storage protein, hexamerin, which acts as a suppressor of soldier formation, is largely synthesized inside the fat body and constitutes a significant proportion of the protein in Reticulitermes flavipes (Kollar) (Zhou, Oi & Scharf, 2006). A series of metabolic pathways, including carbohydrate, amino acid, and lipid metabolism pathways, are also mobilized to produce the metabolites needed for the transformation to soldier in Coptotermes formosanus Shiraki (Du et al., 2020). Previous gene expression studies have shown that the signaling pathways, especially those related to JH, such as the JH signaling pathway (Masuoka et al., 2015), insulin signaling pathway (Hattori et al., 2013), and transforming growth factor β (TGFβ) signaling pathway (Masuoka et al., 2018), participate in soldier caste differentiation. In one previous study, the JH and insulin signaling pathways stimulated the expression of the Hox gene deformed, which further stimulated the appendage-patterning gene dachshund, influencing the elongation of soldier mandible in H. sjostedti (Sugime et al., 2019). The transformation from worker to presoldier is regulated by a complex network (Korb & Hartfelder, 2008; Miura & Scharf, 2011; Miura & Maekawa, 2020), but the regulation of soldier caste differentiation has mostly been studied at the transcriptional level.

MicroRNAs (miRNAs) play an important role in post-transcriptional regulation through mRNA degradation and translation inhibition (Huntzinger & Izaurralde, 2011). miRNAs only need an ∼7-nt seed sequence to target mRNAs and are thus widely engaged in the post-transcriptional gene expression regulation. miRNAs regulate a wide range of physiological activities, such as development, reproduction, and immunity in insects (Belles, 2017; Lucas et al., 2015; Song & Zhou, 2020). Previous studies have reported that miRNAs are involved in caste regulation in insects. For example, the expression levels of miR-6001-5p and -3p are higher in queen-destined larvae than in worker-destined larvae of Apis mellifera L., indicating these miRNAs may be involved in caste differentiation (Collins et al., 2017). In A. mellifera, miRNA-34 and miRNA-210 may affect the brain development of honeybee castes (Vieira et al., 2021), and miR162a has been shown to promote the formation of worker bees by regulating the target of rapamycin (TOR) (Zhu et al., 2017). Although miRNAs can target the JH signaling pathway, insulin signaling pathway, and TGFβ signaling pathway (Ebrahimi et al., 2019; Lozano, Montañez & Belles, 2015; Suzuki, 2018), and these signaling pathways are involved in soldier caste differentiation in termites (Hattori et al., 2013; Masuoka et al., 2015, 2018), experimental proof is still lacking that miRNAs can control soldier caste differentiation by targeting these pathways.

Considering the widespread role of miRNAs, it is hypothesized that miRNAs regulate soldier caste differentiation. miRNAs may add another layer of gene expression regulation to control soldier caste formation. However, research into the role of miRNAs on caste regulation in termites is limited (Itakura, Hattori & Umezawa, 2018; Matsunami et al., 2019). The objective of this study was to identify the miRNAs involved in soldier caste differentiation in C. formosanus.

Materials and Methods

Methoprene bioassay

Termites (C. formosanus) were collected using a termite ground trap in Shangchong Fruit Tree Park, Guangzhou, China (Su & Scheffrahn, 1986; Chouvenc, Ban & Su, 2022). Only workers and soldiers were collected with the trap. The termites were then kept at room temperature and were used within two weeks of collection.

Soldiers were artificially induced using a JHA (methoprene) as follows. Filter paper (32 mm in diameter) was treated with acetone solution of methoprene and air dried for 15 min to evaporate the acetone, making the concentration of methoprene 1,000 ppm. The acetone-treated filter paper was used as a control. Two pieces of treated filter paper were placed in a plastic Petri dish (35 mm in diameter). A volume of 245 μL water was added to the two pieces of filter paper. Twenty workers were then put into the Petri dish. One colony of termites was used in this study. There were three replicates each in the methoprene-treated filter paper group (M group) and acetone-treated filter paper group (C group). After feeding on the filter paper for four days, the workers were collected and decapitated with a scalpel. The worker heads from the same Petri dish were placed into an Eppendorf tube with 100 μL Trizol reagent. The tube was then immersed in liquid nitrogen. All the samples were stored at −80 °C before use.

RNA extraction and sequencing

Total RNA was extracted using TRIzol reagent. The concentration and integrity of the RNA were measured using an Agilent 2100 Bioanalyzer (Agilent Technologies, Palo Alto, CA, USA). RNAs of 18–30 nt were enriched by polyacrylamide gel electrophoresis. Six libraries (three for the M group, three for the C group) were constructed by the NEBNext small RNA library prep set (New England Biolabs; E7300). Briefly, the 3’ and 5’ adapters were added to the RNAs. Then, the RNAs were reverse transcribed and amplified by PCR. The PCR products (140–160 bp in length) were recycled for sequencing. The libraries were evaluated by the Agilent 2100 and the ABI StepOnePlus Real-Time PCR System (Life Technologies, CA, United States). Finally, the libraries were sequenced by Illumina Hiseq2000 (Illumina, San Diego, CA, USA).

Identification of miRNAs and miRNA target prediction

The sequence of the adapters and the low-quality bases were filtered out to acquire clean tags. The following reads were filtered out: reads containing more than one low-quality (Q-value ≤ 20) or unknown nucleotides; reads without 5’ or 3’ adapters; reads without a nucleotide between the 5’ and 3’ adapters; reads containing polyA; and reads shorter than 18 nt (not including adapters). The clean reads were then aligned with small RNAs in GeneBank and Rfam using Blastall 2.2.25 (blastn). The reads of the non-coding RNAs (ncRNAs), including rRNAs, scRNAs, snoRNAs, snRNAs, and tRNAs, were removed from the clean tags. The miRNAs of C. formosanus were not included in the miRBases, so the remaining tags were searched against the miRBases database to identify known miRNAs by aligning with the miRNAs of other species using Bowtie (version 1.1.2) (Langmead & Salzberg, 2012). For the known miRNA of C. formosanus, x denoted that the miRNA was aligned with a 5’ mature miRNA in the miRBases database; y denoted that the miRNA was aligned with a 3’ mature miRNA in the miRBases database. Possible novel miRNAs were identified according to their position on the unigenes and hairpin structures predicted by Mireap_v0.2 using the default parameters for animals. The reads of the known miRNAs and novel miRNAs were then removed from the clean tags, and the remaining tags were aligned with the RNA-Seq data of C. formosanus to remove the reads from mRNA degradation. The RNA-Seq data were obtained by sequencing the transcriptomes of C. formosanus workers feeding on methoprene-treated filter paper and workers feeding on acetone-treated filter paper for four days (Du et al., 2020). After identification of ncRNAs, known miRNAs, novel miRNAs, and reads from mRNA degradation, the rest of the tags were recorded as unannotated.

The transcriptomes of C. formosanus were used for miRNA target prediction (Du et al., 2020). The target genes of miRNAs were predicted by RNAhybrid (v2.1.2) + svm_light (v6.01), Miranda (v3.3a), and TargetScan (Version:7.0). The intersection of the target genes identified by all three of these methods were identified as the miRNA target genes.

Analysis of sample relationship and differentially expressed miRNAs (DEmiRs)

The expression levels of miRNAs were normalized to transcripts per million (TPM). The normalized expression levels of miRNAs were subjected to a principal component analysis (PCA) using the R package gmodels (R Core Team, 2013). The software edgeR was used to identify the differentially expressed miRNAs between the C group and the M group with the criteria |log2(fold change)| > 1 and P value < 0.05 (Robinson, McCarthy & Smyth, 2010). The statistical power analysis was performed using RNASeqPower.

Validation of miRNA sequencing results

The expression levels of the miRNAs were quantified using the methods outlined by Chen et al. (2005). RNA samples were reverse transcribed using stem-loop RT primers and the cDNAs were analyzed by dye-based qPCR. The stem-loop RT primers for the target genes and the reference gene were used to reverse transcribe the RNA samples with M-MuLV (NEB, Ipswich, Massachusetts, US). The PowerUp SYBR Green Master Mix (Thermo Fisher Scientific, Waltham, MA, USA) was used to conduct the qPCR with the miRNA specific forward primer and a universal reverse primer (sequence: GTGCAGGGTCCGAGGT). Four miRNAs were selected from the DEmiRs, i.e., let-7-y, miR-1175-y, miR-181-x, and miR-305-y. Novel-m0649-y was used as the reference gene based on the screening results of the most stable reference gene (Du et al., 2023). The sequences of the stem-loop RT primers and the forward primers are summarized in Table S1. The data were analyzed using the 2−ΔΔCt method (Livak & Schmittgen, 2001). The 2−ΔΔCt data were log2 transformed and analyzed using a t-test to compare the expression value between the C group and the M group.

Determination of DEmiR-regulated genes

The differentially expressed genes (DEGs) were obtained from the transcriptome data of C. formosanus (Du et al., 2020). Because miRNAs mediate the repression of mRNA expression post-transcriptionally, the DEmiRs with the potential to control soldier caste differentiation should have an inverse expression relationship with their targeting DEGs. For visualization of the DEmiR-DEG regulatory network, a pair-wise Spearman correlation was conducted for DEmiR-DEG targeting pairs using R. The DEmiR-DEG targeting pairs with Spearman’s rank correlation coefficient <−0.6 and P < 0.05 were selected to construct the DEmiR-DEG regulatory network with Cytoscape. For clarity, only the top 18 most expressed DEmiRs were shown from a total of 116 DEmiRs.

Enrichment analysis of the DEmiR-targeted DEGs

Gene Ontology (GO) systematically describes the features of genes and gene products with a standardized vocabulary in three categories: cellular components, molecular functions, and biological processes. The Kyoto Encyclopedia of Genes and Genomes (KEGG) is a database incorporating molecular-level data to assist in the exploration of high-level functions and utilities of biological systems. KEGG pathway establishes a relatively comprehensive group of pathway maps about interaction, reaction, and relation networks at the molecular level. In order to evaluate the functions of the DEmiRs in soldier caste differentiation and identify the significant GO terms and KEGG pathways that are regulated by miRNAs, the DEmiR-regulated DEGs were subjected to the GO and KEGG pathway enrichment analysis using Fisher’s exact test. False discovery rate was used to adjust the P value.

Results

Sequencing quality

Six libraries were constructed, three for each treatment. The sequencing quality of the libraries is shown in Table S2. There were 74,791,183 clean reads after filtering out the invalid sequences. After removing the low-quality reads and contaminants, including adapters, inserts, polyA reads, reads smaller than 18 nt, and low cutoff reads, a total number of 67,959,005 clean tags were acquired. There were 33,598,522 and 34,360,483 clean tags for the three libraries in the C group and the M group, respectively (Tables 1 and S2). All the samples had more than 91% clean tags. The length distribution of the tag is a characteristic animal small RNA, with the highest percentage at 22 nt. The percentage of the 22 nt tag was 33.93%, 33.91%, 34.41%, 34.29%, 33.37%, 34.97% for C-1, C-2, C-3, M-1, M-2, M-3, respectively (Fig. S1).

Table 1 Read counts and percentage of different tags in the C group and the M group.

Type	C	M	
Read count	Percentage (%)	Read count	Percentage (%)	
rRNA	106,350 ± 5,708	0.95 ± 0.03	126,087 ± 22,500	1.12 ± 0.25	
snRNA	4,176 ± 1,352	0.04 ± 0.01	4,518 ± 2,550	0.04 ± 0.02	
snoRNA	378 ± 34	0.003 ± 0.000	135 ± 24	0.001 ± 0.000	
tRNA	44,440 ± 10,750	0.40 ± 0.10	32,962 ± 11,163	0.30 ± 0.11	
Known miRNA	8,569,615 ± 167,997	76.52 ± 0.47	8,881,603 ± 723,339	77.37 ± 1.93	
Novel miRNA	43,863 ± 2,129	0.39 ± 0.01	42,076 ± 2,786	0.37 ± 0.00	
Transcriptome	763,489 ± 27,327	6.82 ± 0.19	714,109 ± 35,751	6.24 ± 0.08	
Unannotated	1,667,196 ± 45,213	14.89 ± 0.29	1,652,004 ± 115,416	14.56 ± 1.52	
Total	33,598,522	100	34,360,483	100	
Note:

C = control group, M = methoprene-treated group. Workers in the C group were fed with acetone-treated filter paper; workers in the M group were fed with methoprene-treated filter paper.

Identification of miRNAs and target prediction

Four ncRNAs, including rRNAs, snRNAs, snoRNAs, and tRNAs, were identified by searching the Rfam ncRNA database, while only two ncRNAs (rRNAs and tRNAs) were identified using the Genbank ncRNA database. However, more counts of rRNAs and tRNAs were identified using Genbank than when using Rfam (Table S3). The percentage of known miRNAs was the highest among all the tags in either the M or the C group, and its percentage in the C group was similar to that in the M group (76.52 ± 0.47% vs 77.37 ± 1.93%; Table 1). However, the reads annotated as novel miRNAs only accounted for a small portion of the total reads (0.39 ± 0.01% in the C group, 0.37 ± 0.00% in the M group). There were 6.82 ± 0.19% counts which were identified as transcriptome data in the C group, and 6.24 ± 0.08% in the M group. A total of 1,324 miRNAs were identified, with 589 known miRNAs and 735 novel miRNAs (Table 2). In addition, 21,281 genes were predicted as the targets of the miRNAs. The miRNAs and their computationally-predicted-targets are summarized in Table S4.

Table 2 Number of miRNAs, target genes, and target sites identified.

Sample	Known miRNA	Novel miRNA	Total miRNA	Target gene	Target site	
C	373 ± 32	555 ± 5	928 ± 36	20,852 ± 47	1,007,981 ± 31,277	
M	315 ± 25	543 ± 5	858 ± 20	20,700 ± 20	924,439 ± 9,607	
Total	589	735	1,324	21,281	1,405,633	
Note:

C = control group, M = methoprene-treated group. Workers in the C group were fed with acetone-treated filter paper; workers in the M group were fed with methoprene-treated filter paper.

Sample relationship between treatments

The PCA analysis showed high variability among samples (Fig. 1). The first component only accounted for 35.6% of the variance, while the second component accounted for 20.2% of the variance. According to the 3-dimensional figure of the PCA analysis, the third component accounted for 17.3% of the variance. The three components together accounted for 73.1% of the total variance.

Figure 1 Principal component analysis (PCA) of the expression levels of miRNAs.

PCA analysis showed that replicates from the same treatment were more likely to cluster together. C1-C3 are the three replicates from the control group (C) and M1-M3 are the three replicates from the methoprene-treated group (M).

DEmiRs between treatments

There were 116 differentially expressed miRNAs between the C group and the M group: 16 up-regulated miRNAs and 100 down-regulated miRNAs in the M group, using the C group as the reference. Some of the DEmiRs were expressed at relatively low expression levels. The DEmiRs, their sequences, and their expression levels are shown in Table S5. The statistical power of this experimental design was 0.41 (n = 3).

Validation of miRNA sequencing results

The qPCR data were only partially consistent with the miRNome data. There were significant differences between the C group and the M group for let-7-y and miR-305-y (Figs. 2A and 2B). According to the qPCR results, there was no significant difference for miR-1175-y and miR-181-x between the C group and the M group (Figs. 2C and 2D), but the miRNome sequencing data showed different expression levels of miR-1175-y and miR-181-x between the C group and the M group.

Figure 2 Verification of DEmiR expression levels through qPCR.

Presented in the graph are the expression levels of (A) let-7-y, (B) miR-305-y, (C) miR-1175-y, and (D) miR-181-x in the C group and the M group. The left y-axis is the expression level of the gene determined by qPCR (black histogram), and the right y-axis is the expression level of the gene acquired by miRNA sequencing (grey histogram). The data are presented as mean ± SE. NS means no significant difference. One asterisk (*) and two asterisks (**) mean significant difference at P < 0.05 and P < 0.01, respectively. C = control group, M = methoprene-treated group.

DEmiR-DEG pairs and functional analysis of the DEmiR-regulated DEGs

A total of 2,547 DEGs were found between the C group and the M group after treating workers with methoprene for four days, including 1,480 up-regulated genes, and 1,067 down-regulated genes in the M group, using the C group as the reference (Du et al., 2020). In combination with the results of the inverse relationship and the targeting relationship, a total of 4,433 DEmiR-DEG pairs were obtained (Table S6). The DEGs in the DEmiR-DEG pairs were considered to be the genes regulated by the DEmiRs. Among the top 18 most expressed DEmiRs, miR-148-y targeted the most DEGs that were down-regulated after workers were treated with methoprene (Fig. 3A), while let-7-y targeted the largest number of genes that were up-regulated after methoprene treatment (Fig. 3B). The KEGG pathway classification and GO annotation of the DEmiR-targeted DEGs are shown in Table S6. The DEmiRs were involved in the regulation of genes in metabolic pathways, the TGFβ signaling pathway, hedgehog signaling pathway, and FoxO signaling pathway (Table S6).

Figure 3 MicroRNA-mRNA regulatory networks during solider differentiation in C. formosanus.

(A) Down-regulated miRNAs after workers were treated with methoprene and their target genes. (B) Up-regulated miRNAs after workers were treated with methoprene and their target genes.

Although several GO items in the cellular component, molecular function, and biological process categories were enriched according to the P values, no GO term was recognized as significant after adjusting the P values (Fig. 4). The KEGG enrichment analysis showed that there were two significantly enriched pathways: ribosome biogenesis in eukaryotes (Q < 0.009) and the circadian rhythm-fly pathway (Q < 0.036; Fig. 5). The top 10 enriched pathways were mainly related to biogenesis and metabolism, with the circadian rhythm-fly pathway being the only enriched KEGG pathway related to signal transduction.

Figure 4 The GO analysis of the DEmiR-targeted DEGs.

The top 20 enriched GO terms are shown in the bubble chart. No GO terms were enriched according to the GO analysis. Note: The y-axis is the name of the GO term, the x-axis is the rich factor.

Figure 5 The KEGG enrichment analysis of the DEmiR-targeted DEGs.

The top 20 enriched KEGG pathways are shown in the bubble chart. Note: the y-axis is the name of the pathway, the x-axis is the rich factor.

Discussion

Through the sequencing of the miRNomes of workers of C. formosanus, a total of 1,324 miRNAs were identified. High variability in miRNA expression levels among the three replicates was observed, because the first component only explained 35.6% of the variance, as indicated by the PCA analysis. The variability among the replicates was also demonstrated by high standard errors of the relative expression values of the genes seen in the qPCR results. A total of 116 candidate DEmiRs were identified that have an inhibitory effect on the DEGs. However, many DEmiRs had very low expression levels, which may not be detected by the qPCR method. Most of the low-expression DEmiRs were novel miRNAs, so there is a high likelihood that they do not play an important role in soldier caste regulation. Most of the DEmiRs with high expression levels were conserved miRNAs, so the differentiation of soldier caste is more likely to be regulated by conserved miRNAs (Matsunami et al., 2019). Because a seed sequence of only ∼7-nt long is required for mRNA target recognition, a miRNA has many target genes, and a gene can be regulated by many miRNAs. Thus, the DEmiRs and DEGs form a complex network regulating soldier caste differentiation.

JH and JH signaling pathways play an essential role in caste differentiation. Knockdown of the JH receptor gene Met affects the morphogenesis of the soldier mandible (Masuoka et al., 2015). The JH signaling pathway is regulated by miRNAs. The miR-2 family (miR-2, miR-13a, and miR-13b) targets Kr-h1 (the downstream gene of Met) in Blattella germanica (L.), facilitating Kr-h1 degradation and thus promoting metamorphosis from nymphs to adults (Lozano, Montañez & Belles, 2015). let-7 and miR-278 regulate Kr-h1 to control metamorphosis and oogenesis in Locusta migratoria L. (Song et al., 2018), but in this study, no target site of let-7-y was found in the 3’ UTR of Kr-h1 of C. formosanus. The miRNA target site is under selection pressure, and dynamic changes of the miRNA target site have been observed (Xu et al., 2013). This is likely one reason for the loss of the let-7-y target site in Kr-h1 in C. formosanus. let-7, one of the first discovered miRNAs, is conserved across species, and plays an important role in the regulation of cell differentiation and proliferation (Roush & Slack, 2008). According to the miRNome results of C. formosanus, 35 genes were found to have a target relationship and negative correlation relationship with let-7-y, including cytochrome P450, which has been reported to be involved in soldier caste differentiation (Tarver, Coy & Scharf, 2012). Therefore, there is a high possibility that let-7-y is engaged in soldier caste differentiation.

The KEGG enrichment analysis of the DEmiR-targeted DEGs revealed two enriched KEGG pathways, one of which was the circadian rhythm-fly pathway. There are many control genes for circadian rhythm, and period was the first discovered (Bargiello, Jackson & Young, 1984; Zehring et al., 1984). The regulation of period includes a feedback mechanism. The transcription of period is activated by the CLOCK-BMAL1 heterodimers (Gekakis et al., 1998), and PER can inhibit its own transcription by inhibiting CLOCK activity (Darlington et al., 1998). The circadian rhythm genes can also control other non-circadian-rhythm activities, such as cancer development (Wood, Yang & Hrushesky, 2009). The circadian rhythm-fly pathway was enriched in the KEGG enrichment analysis after treating workers with methoprene in C. formosanus, indicating the circadian rhythm-fly pathway may regulate soldier differentiation (Du et al., 2020). period is one of the differentially expressed genes in the circadian rhythm-fly pathway in C. formosanus. After checking the DEmiR-DEG pairs in C. formosanus, 23 miRNAs with expression levels that were negatively correlated with period were identified, including miR-142-x, miR-30-x, and miR-21-x, indicating that these miRNAs may be involved in soldier caste differentiation by regulating the circadian rhythm-fly pathway.

The insulin signaling pathway is also involved in soldier caste differentiation (Hattori et al., 2013; Sugime et al., 2019). The FoxO pathway has crosstalk with the insulin signaling pathway (Accili & Arden, 2004). FoxO has been found to be differentially expressed after treating workers with JHA (Du et al., 2020; Hattori et al., 2013), indicating that the FoxO pathway may also be involved in soldier caste differentiation. FoxO is regulated by many miRNAs, such as miR-9, miR-10b, and miR-21 (Urbánek & Klotz, 2017). The miRNAs which targeted and anti-correlated with FoxO include miR-144-y and miR-19-y in C. formosansus. The downstream gene of FoxO, BNIP3, is also regulated by miRNA (Chen et al., 2010). BNIP3 formed anti-correlation DEmiR-DEG pairs with miR-142-x and miR-181-x during soldier differentiation in C. formosansus. According to the miRNome data in this study, miRNAs may play an important role in soldier caste differentiation by regulating the FoxO pathway.

Besides being involved in signaling transduction regulation, miRNAs have also been reported to control metabolic pathways. For example, miRNA-133 regulates dopamine synthesis to control the density-dependent phenotype in locusts (Yang et al., 2014). Of the top 10 enriched pathways in the present study, nine of them were related to biogenesis and metabolism. Ribosome biogenesis in eukaryotes was the most of the enriched pathway in this study. The ribosome is the site for protein translation and ribosome biogenesis is related to cell proliferation. During soldier caste differentiation, fat body increases significantly (Cornette, Matsumoto & Miura, 2007). Correspondingly, many proteins, such as hexamerin, are largely synthesized (Zhou, Oi & Scharf, 2006). This may explain the importance of the ribosome biosynthesis in soldier caste differentiation. miRNAs are involved in the regulation of ribosome biogenesis (McCool, Bryant & Baserga, 2020). According to the miRNome data in this study, 11 DEGs related to ribosome biosynthesis were regulated by 45 DEmiRs, indicating miRNAs form a complex network to regulate ribosome biosynthesis, thus regulating soldier caste differentiation.

Termite colonies are able to maintain a stable ratio of soldiers, indicating that the transformation of worker to soldier is precisely controlled in termites. The roles of miRNAs in controlling caste differentiation in the honey bee have been widely reported (Ashby et al., 2016; Guo et al., 2016; Guo et al., 2013; Shi et al., 2015). This study identified a candidate pool of miRNAs which may be involved in soldier caste regulation in termites. The regulation of miRNAs in soldier caste differentiation adds another layer to soldier caste differentiation to improve the precision of the process. Further research is needed to study the function of these miRNAs and confirm the target relationships with corresponding mRNAs.

Conclusions

miRNAs are powerful regulators of various cellular activities, such as differentiation, proliferation, and development (Inui, Martello & Piccolo, 2010; Plasterk, 2006). This study aimed to investigate the functions of miRNAs in the soldier caste differentiation of termites. By comparing the miRNomes of workers feeding on methoprene-treated filter paper and workers feeding on acetone-treated filter paper, 116 DEmiRs were identified between the two groups. These DEmiRs formed a complex network with DEGs, including genes in ribosome biogenesis, the circadian rhythm-fly, and metabolic pathways. By controlling soldier caste differentiation post-transcriptionally, miRNAs add another layer of gene regulation to maintain a stable soldier ratio in the termite colony, thus maintaining colony homeostasis.

Supplemental Information

Supplemental Information 1 Raw data for qPCR.

Supplemental Information 2 The length distribution of the tags in each library.

(A)–(F) are the length distribution of the tags from C-1, C-2, C-3, M-1, M-2, M-3, respectively. C1–C3 are the three replicates from the control group (C) and M1–M3 are the three replicates from the methoprene-treated group (M).

Supplemental Information 3 Stem-loop RT primers and forward primers for verification of miRNA sequencing results.

Supplemental Information 4 Counts number and frequency of different classes of reads in each snRNA library.

Supplemental Information 5 Read counts and percentage of non-coding RNA matched with those in Genbank and Rfam.

Supplemental Information 6 The miRNAs and their computationally-predicted targets.

Supplemental Information 7 The expression profiles of miRNA in the six libraries.

Supplemental Information 8 List of the DEmiR-DEG pairs.

Additional Information and Declarations

Competing Interests

Author Contributions

DNA Deposition

Data Availability

The authors declare that they have no competing interests.

He Du conceived and designed the experiments, analyzed the data, prepared figures and/or tables, authored or reviewed drafts of the article, and approved the final draft.

Runmei Huang performed the experiments, authored or reviewed drafts of the article, and approved the final draft.

Da-Song Chen analyzed the data, authored or reviewed drafts of the article, and approved the final draft.

Tianyong Zhuang performed the experiments, authored or reviewed drafts of the article, and approved the final draft.

Xueyi Huang performed the experiments, authored or reviewed drafts of the article, and approved the final draft.

Huan Zhang analyzed the data, authored or reviewed drafts of the article, and approved the final draft.

Zhiqiang Li analyzed the data, authored or reviewed drafts of the article, and approved the final draft.

The following information was supplied regarding the deposition of DNA sequences:

The RNA-seq data are available at the GenBank Sequence Read Archive: PRJNA577439.

The miRNome data are available at GenBank Sequence Read Archive: PRJNA922245.

The following information was supplied regarding data availability:

The raw data for qPCR are available in the Supplemental Files.

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
