# Peer review of "Regulation of soldier caste differentiation by microRNAs in Formosan subterranean termite (Coptotermes formosanus Shiraki)"

_PeerJ, doi:10.7717/peerj.16843_

## Round 0.1 · original submission · Major Revisions

Both authors have appreciated your work. They also underline several points needing your attention.

Reviewer 1 ·

Basic reporting

Du et al. analyzed microRNAs whose expression changes after a methoprane application in the Formosan subterranean termites. Since the methoprane application causes soldier differentiation in this species, it is possible that this study captures the changes in microRNA expression that occur early in soldier differentiation. There is no major flaw in the overall research.

Experimental design

There is no problem with the experimental design.

Validity of the findings

The content could be useful to the community in this research field.

Additional comments

Some comments are provided below.

In the Introduction, in many cases, it is not written about which species the authors are mentioning. There is no guarantee that the phenomenon is common to all termite species.

Line 55, lip -> lipid

Line 71, the authors should write clearly which insect they are mentioning. Here it is bumblebee.

Line 93, What does this indicate is 1000 ppm of mesoprene to the weight of what? Is it the weight of the filter paper? Is the mesoprene dissolved in acetone and soaked into the filter paper?

In the Discussion, I think authors should state why the qPCR results do not match the RNA-seq results. Which is more reliable, RNA-seq or qPCR? I think this should be handled with caution, as it relates to the credibility of the data in this research article as a whole.

Lines 278-293, notation of a gene in Drosophila should begin with a lowercase letter if it is found in a recessive phenotype. period as a gene or transcript should begin with a lowercase letter if the authors follow the notation in Drosophila.

Line 296, FoXO -> FoxO. This gene is conventionally written as foxo, FoxO, or FOXO, but at least in this article it is better to use one of them.

Lines 233 and 322 and Fig 5. I could not find the pathway called "tMetabolic pathway" in KEGG database. Are they typos?

Reviewer 2 ·

Basic reporting

The submitted manuscript is written in a concise way, using proper style and language. It has a professional structure and the amount of references is appropriate.

Experimental design

There are some issues that needs to be reported regarding the experimental design to improve quality, the details are provided in "additional comments" section

Validity of the findings

This paper is interesting and has a significant scientific value. Conclusions, based on the comprehensive literature and analysis, are linked to the results.

Additional comments

General comment: Along the text, in several cases, the authors include abbreviations that are not common sense without explanation, such as TGFß, amTOR, FDR adjust. Please include in parenthesis the complete name when they appear for the first time

Abstract: Please remove the subheadings (background, methods, results)
Add the expected function of Methoprene (i.e. mimics juvenile hormone or interfere in growth regulation).
At the end of the abstract, include how miRNA are involved in the regulatory network, as you mentioned in lines 317-319

Introduction:
Line 63: you state that “The transformation from worker to presoldier is regulated by a complex network”. Please add references to corroborate this.
Lines 71-72: adjust the phrase for clarification; the correct expression in this context is ‘highly expressed’.
Line 80: microRNA or miRNA? Please standardize the term and adjust the phrase to avoid repetition
Line 84: “The objective of this project is to identify”… Please adjust the term ‘Project’ and put the phrase at the past.

Material and methods:
Please add a sentence about the biology of the termite used in this assay and how many castes they can differentiate, especially whether they have one or more types of soldiers.
Line 89: How many colonies were collected?
Line 97-98: the authors said “The workers were collected 4 days after feeding”, but they not mentioned whether the termites were fed at the first day at the plates or later. Please clarify how many days the termites were exposed to each treatment.
Line 109: mention how many libraries were sequenced
Line 123-124: Mention the reference of the RNAseq data or deposit number
Line 130: please clarify what is x and y
Line 160: please explain why did you “consider only the inhibitory effect of miRNA on the gene expression” even though you state that “there are reports that miRNA could up-regulate the gene expression level”
Line 166: please add the total number of DEmiRs at the end of the sentence, as follows: For clarity, only the top 18 most expressed DEmiRs was shown from ___.

Results
Lines 202 and 204: please report the exact value of p; not the threshold of 0.05.
Line 221: Add a sentence highlighting that TMP were different from C and M in Fig2C and Fig2D.
Figure 3: If possible, change the color of each cluster to improve clarity, especially in A.

Discussion
Lines 249-251 should be included in Material and methods
Line 256: please add a sentence explaining why
Lines 270-271: The structure of this sentence seems that the method of analysis is inaccurate. Please rephrase and explain it better.
Lines 279-286: Please rephrase to connect the genes described here to ribosome biogenesis and circadian rhythm-fly. The information is added in next sentences but the text is not cohesive.

---

## Round 0.2 · Minor Revisions

Thank you for the improvements introduced in the revision of the manuscript. As Reviewer 2 indicates, the manuscript presents interesting results, but it still needs your attention concerning some minor aspects.

Reviewer 2 ·

Basic reporting

The submitted manuscript improved the language, style, and references provided.

Experimental design

I was concerned about the collection of only one colony, but the discussion has a proper tone for the inferences they can do regarding the results

Validity of the findings

This paper is interesting and has a significant scientific value.

Additional comments

I would like to thank the authors for the effort to address all comments I had at the first version. Abstract is adjusted accordingly and there is some few adjusts needed at this point along the manuscript:
41: ‘soldier’ is misspelled, please check through the text
41: This sentence is unclear. Perhaps change ‘a termite’ to ‘each termite’ species
45: ‘soldier’ is misspelled
Figure 2: Please improve legend and explain which treatments are shown in a, b, c, or d. The text is better but the legend is vague
Figure 3: ‘soldier’ is misspelled at the legend
259-260: when you state that “Most of the DEmiRNAs with high expression levels were conserved miRNAs”, please add a reference

---

## Round 0.3 · accepted · Accept

I have verified the last version submitted and I thank you very much for having properly addressed the reviewer's comments. In my opinion, the manuscript is ready for publication.